# MCMC Variational Inference via Uncorrected Hamiltonian Annealing

**Tomas Geffner**
College of Information and Computer Science
University of Massachusetts, Amherst
Amherst, MA
tgeffner@cs.umass.edu

**Justin Domke**
College of Information and Computer Science
University of Massachusetts, Amherst
Amherst, MA
domke@cs.umass.edu

## Abstract

Given an unnormalized target distribution we want to obtain approximate samples from it and a tight lower bound on its (log) normalization constant $\log Z$. Annealed Importance Sampling (AIS) with Hamiltonian MCMC is a powerful method that can be used to do this. Its main drawback is that it uses non-differentiable transition kernels, which makes tuning its many parameters hard. We propose a framework to use an AIS-like procedure with Uncorrected Hamiltonian MCMC, called Uncorrected Hamiltonian Annealing. Our method leads to tight and differentiable lower bounds on $\log Z$. We show empirically that our method yields better performances than other competing approaches, and that the ability to tune its parameters using reparameterization gradients may lead to large performance improvements.

## 1 Introduction

Variational Inference (VI) [4, 41, 45] is a method to do approximate inference on a target distribution $p(z) = \bar{p}(z)/Z$ that is only known up to the normalization constant $Z$. The basic insights are, first, that the evidence lower bound (ELBO) $\mathbb{E}_{q(z)}[\log \bar{p}(z) - \log q(z)]$ lower-bounds $\log Z$ and, second, that maximizing the ELBO is equivalent to minimizing the KL-divergence from $q$ to $p$. The simplest VI method chooses a parameterized family for $q$ and optimizes its parameters to maximize the ELBO.

A recent direction involves combining VI with Markov chain Monte Carlo (MCMC) [34, 43]. These methods can be seen as an instance of the auxiliary VI framework [2] – they create an augmented variational distribution that represents all intermediate random variables generated during the MCMC procedure. An augmented target distribution that attempts to capture the inverse MCMC dynamics is optimized jointly with this variational distribution. However, it has been observed that capturing inverse dynamics is challenging [43, §5.4] (further discussion in Section 4).

Annealed Importance Sampling (AIS) [22, 27] is a powerful technique used to build augmented distributions without the need of learning inverse dynamics. While it was originally proposed to estimate expectations using importance sampling, it can be easily used to build lower bounds on normalization constants of intractable densities [18, 44]. AIS creates a sequence of densities that bridge from a tractable initial approximation $q$ to the target $\bar{p}$. Then, the augmented variational distribution is given by a sequence of MCMC kernels targeting each bridging density, while the augmented target uses the *reversals* of those kernels. It turns out that the ratio of these augmented distributions can be computed using only evaluations of the bridging densities. Combining Hamiltonian MCMC kernels with AIS has been observed to produce strong lower bounds [35, 44].

However, these bounds are sensitive to numerous parameters, such as the initial distribution, bridging schedule, and parameters of the MCMC kernels. It would be desirable to optimize these parameters

to tighten the bound. Unfortunately, the presence of Metropolis-Hastings acceptance steps means that the the final estimator is non-differentiable, and thus reparameterization gradients cannot be used.

In this work, we propose *Uncorrected* Hamiltonian Annealing (UHA), a differentiable alternative to Hamiltonian AIS. We define an augmented variational distribution using Hamiltonian MCMC kernels, but dropping the accept-reject steps. This is motivated by the fact that Hamiltonian dynamics sometimes have high acceptance rates. Since these uncorrected MCMC kernels do not exactly hold the bridging densities invariant, an augmented target distribution cannot be defined in terms of reversals. Instead, we define our augmented target by deriving an algorithm for the exact reversal of the original (corrected) MCMC kernel and dropping the accept-reject step. Surprisingly, this yields a very simple expression for the resulting lower bound.

We use reparameterization gradients to tune various parameters involved in the lower bound produced by UHA, including the initial approximation $q$, parameters of the uncorrected MCMC kernel, and the bridging densities. Experimentally, tuning all these leads to large gains. For example, in several inference tasks we observe that tuning UHA with $K = 64$ bridging densities gives better results than traditional Hamiltonian AIS with $K = 512$.

Finally, we use UHA to train VAEs [24, 31]. In this case we observe that using UHA leads to higher ELBOs. In addition, we observe that increasing the number of bridging densities with UHA consistently leads to better results, and that for a large enough number of bridging densities the variational gap (difference between ELBO and true log-likelihood) becomes small, and models with higher log-likelihood are obtained.

## 2  Preliminaries

**Variational inference and augmentation.** Suppose that $p(z) = \frac{1}{Z}\bar{p}(z)$ is some target density, where $\bar{p}$ is unnormalized and $Z = \int \bar{p}(z)dz$ is the corresponding normalizer, and let

$$\text{ELBO}(q(z), \bar{p}(z)) = \mathop{\mathbb{E}}_{q(z)} \log \frac{\bar{p}(z)}{q(z)} \tag{1}$$

be the "ELBO operator". Variational inference (VI) is based on the fact that for any $q(z)$ we have [4]

$$\log Z = \text{ELBO}(q(z), \bar{p}(z)) + \text{KL}(q(z)\|p(z)). \tag{2}$$

In VI, the parameters of $q$ are tuned to maximize the "evidence lower bound" (ELBO). Since the KL-divergence is non-negative, this is always a lower bound on $\log Z$. Also, maximizing the ELBO is equivalent to minimizing the KL-divergence from $q$ to $p$.

To get tighter bounds and better approximations recent work has made use of *augmented* distributions [2, 21]. Let $z_{1:M} = (z_1, \cdots, z_M)$ and suppose that $\bar{p}(z_{1:M}) = \bar{p}(z_M)p(z_{1:M-1}|z_M)$ augments the original target density while preserving its normalization constant. Then, for any $q(z_{1:M})$ we have

$$\log Z = \text{ELBO}(q(z_{1:M}), \bar{p}(z_{1:M})) + \text{KL}(q(z_{1:M})\|p(z_{1:M})). \tag{3}$$

The first term is called the "augmented" ELBO and again lower bounds $\log Z$. By the chain rule of KL-divergence [12], the KL-divergence from $q$ to $p$ over $z_{1:M}$ upper-bounds the KL-divergence over $z_M$. This justifies using the marginal of $q$ over $z_M$ to approximate the original target distribution.

**Annealed Importance Sampling.** A successful approach for creating augmented distributions is Annealed Importance Sampling (AIS) [27]. It creates an augmented proposal distribution $q$ by applying a sequence of transition densities $T_m(z_{m+1}|z_m)$, and an augmented target by defining transition densities $U_m(z_m|z_{m+1})$. This gives the augmented densities

$$q(z_{1:M}) = q(z_1) \prod_{m=1}^{M-1} T_m(z_{m+1}|z_m) \quad \text{and} \quad \bar{p}(z_{1:M}) = \bar{p}(z_M) \prod_{m=1}^{M-1} U_m(z_m|z_{m+1}). \tag{4}$$

Naively, the ratio of these densities is

$$\frac{\bar{p}(z_{1:M})}{q(z_{1:M})} = \frac{\bar{p}(z_M)}{q(z_1)} \prod_{m=1}^{M-1} \frac{U_m(z_m|z_{m+1})}{T_m(z_{m+1}|z_m)}. \tag{5}$$

To define the transitions $T_m$ and $U_m$, AIS creates a sequence of unnormalized densities $\bar{\pi}_1, \ldots, \bar{\pi}_{M-1}$ that "bridge" from a starting distribution $q$ to the target $\bar{p}$, meaning that $\bar{\pi}_1$ is close to $q$ and $\bar{\pi}_{M-1}$ is close to $\bar{p}$. Then, for each intermediate distribution, $T_m(z_{m+1}|z_m)$ is chosen to be a Markov kernel that holds $\pi_m$ invariant, and $U_m$ to be the reversal of $T_m$ with respect to $\pi_m$, defined as

$$U_m(z_m|z_{m+1}) = T(z_{m+1}|z_m)\frac{\pi_m(z_m)}{\pi_m(z_{m+1})}. \tag{6}$$

This choice produces a simplification so that eq. 5 becomes

$$\frac{\bar{p}(z_{1:M})}{q(z_{1:M})} = \frac{\bar{p}(z_M)}{q(z_1)} \prod_{m=1}^{M-1} \frac{\bar{\pi}_m(z_m)}{\bar{\pi}_m(z_{m+1})}. \tag{7}$$

This can be easily evaluated without needing to evaluate the transition densities. The ratio from eq. 7 can be used to get an expression for the lower bound $\text{ELBO}(q(z_{1:M}), \bar{p}(z_{1:M}))$. Research has shown that the AIS augmentation may lead to extremely tight lower bounds [18, 17, 35, 44].

**Hamiltonian Dynamics.** Many MCMC methods used to sample from $p(z)$ are based on Hamiltonian dynamics [3, 8, 29, 42]. The idea is to create an augmented distribution $p(z, \rho) = p(z)S(\rho)$, where $S(\rho)$ is a distribution over a momentum variable $\rho$ (e.g. a Multivariate Gaussian). Then, one can define numerical integration schemes where $z$ and $\rho$ evolve while nearly holding $p(z, \rho)$ constant. When corrected by a Metropolis-Hastings acceptance step, this can be made to exactly hold $p(z, \rho)$ invariant. This is alternated with a scheme that resamples the momentum $\rho$ while holding $S(\rho)$ invariant. When Hamiltonian dynamics work well, $z$ can quickly move around, suppressing random-walk behavior.

There are a variety of different Hamiltonian MCMC methods, corresponding to different integration schemes, momentum distributions, and ways of resampling the momentum. For instance, HMC and Langevin dynamics use the leapfrog integrator, a Gaussian for the momentum variables and a full resampling of the momentum variables at each step [29, 42]. On the other hand, if the momentum variables are only partially resampled, the under-damped variants of HMC and Langevin dynamics are recovered [29]. It was observed that partial resampling may lead to improved perfomance [9].

It is easy to integrate Hamiltonian dynamics into AIS. First, define an augmented target $\bar{p}(z, \rho) = \bar{p}(z)S(\rho)$ and an augmented starting distribution $q(z, \rho) = q(z)S(\rho)$. Then, create a series of augmented densities $\bar{\pi}_1(z, \rho), \ldots, \bar{\pi}_{M-1}(z, \rho)$ bridging the two as $\bar{\pi}_m(z, \rho) = \bar{\pi}_m(z)S(\rho)$. Finally, define the forward transition $T_m(z_{m+1}, \rho_{m+1}|z_m, \rho_m)$ to be an iteration of a Hamiltonian MCMC method that leaves $\pi_m(z, \rho)$ invariant. We will describe a single transition $T_m$ as a sequence of three steps: (1) resample the momentum; (2) simulate Hamiltonian dynamics and apply an accept-reject step; and (3) negate the momentum. The precise process that defines the transition is shown in Alg. 1. Note that this algorithm is quite general, and compatible with HMC, Langevin dynamics and their underdamped variants (by selecting an appropriate integrator and resampling method).

---

**Algorithm 1** Corrected $T_m(z_{m+1}, \rho_{m+1}|z_m, \rho_m)$

---

1. Sample $\rho'_m$ from some $s(\rho'_m|\rho_m)$ that leaves $S(\rho)$ invariant. Set $z'_m \leftarrow z_m$.
2. Simulate Hamiltonian dynamics as $(z''_m, \rho''_m) \leftarrow \mathcal{T}_m(z'_m, \rho'_m)$.
   Calculate an acceptance probability $\alpha = \min(1, \bar{\pi}_m(z''_m, \rho''_m)/\bar{\pi}_m(z'_m, \rho'_m))$.
   With probability $\alpha$, set $(z'''_m, \rho'''_m) \leftarrow (z''_m, \rho''_m)$. Otherwise, set $(z'''_m, \rho'''_m) \leftarrow (z'_m, \rho'_m)$.
3. Reverse the momentum as $(z_{m+1}, \rho_{m+1}) \leftarrow (z'''_m, -\rho'''_m)$.
   **return** $(z_{m+1}, \rho_{m+1})$

---

Representing $T_m$ this way makes it easy to show it holds the density $\pi_m(z, \rho)$ invariant. The overall strategy is to show that each of the steps 1-3 holds $\pi_m$ invariant, and so does the composition of them [29, §3.2]. For steps 1 and 3 this is trivial, provided that $S(\rho) = S(-\rho)$. For step 2, we require that the simulation $\mathcal{T}_m$ has unit Jacobian and satisfies $\mathcal{T}_m^{-1} = \mathcal{T}_m$. Then, $\mathcal{T}_m$ can be interpreted as a symmetric Metropolis-Hastings proposal, meaning the Metroplis-Hastings acceptance probability $\alpha$ is as given. A typical choice for $\mathcal{T}_m$ that satisfies these requirements is the leapfrog integrator with a momentum reversal at the end. (This reversal then gets "un-reversed" in step 3 for accepted moves.)

Since $T_m$ holds $\pi_m$ invariant, we can define $U_m$ as the reversal of $T_m$ wrt $\pi_m$. Then, eq. 7 becomes

$$\frac{\bar{p}(z_{1:M}, \rho_{1:M})}{q(z_{1:M}, \rho_{1:M})} = \frac{\bar{p}(z_M, \rho_M)}{q(z_1, \rho_1)} \prod_{m=1}^{M-1} \frac{\bar{\pi}_m(z_m, \rho_m)}{\bar{\pi}_m(z_{m+1}, \rho_{m+1})}. \tag{8}$$

Using this ratio we get an expression for the lower bound $\text{ELBO}(q(z_{1:M}, \rho_{1:M}), \bar{p}(z_{1:M}, \rho_{1:M}))$ obtained with Hamiltonian AIS. While this method has been observed to yield strong lower bounds on $\log Z$ [35, 44] (see also Section 5.2), its performance depends on many parameters: initial distribution $q(z)$, momentum distribution $S$, momentum resampling scheme, simulator $\mathcal{T}_m$, and bridging densities. We would like to tune these parameters by maximizing the ELBO using reparameterization-based estimators. However, due to the accept-reject step required by the Hamiltonian MCMC transition, the resulting bound is not differentiable, and thus reparameterization gradients are not available.

## 3 Uncorrected Hamiltonian Annealing

The contribution of this paper is the development of *uncorrected* Hamiltonian Annealing (UHA). This method is similar to Hamiltonian AIS (eq. 8), but yields a differentiable lower bound. The main idea is simple. For any transitions $T_m$ and $U_m$, by the same logic as in eq. 5, we can define the ratio

$$\frac{\bar{p}(z_{1:M}, \rho_{1:M})}{q(z_{1:M}, \rho_{1:M})} = \frac{\bar{p}(z_M, \rho_M)}{q(z_1, \rho_1)} \prod_{m=1}^{M-1} \frac{U_m(z_m, \rho_m | z_{m+1}, \rho_{m+1})}{T_m(z_{m+1}, \rho_{m+1} | z_m, \rho_m)}. \tag{9}$$

Hamiltonian AIS defines $T_m$ as a Hamiltonian MCMC kernel that holds $\pi_m$ invariant, and $U_m$ as the reversal of $T_m$ with respect to $\pi_m$. While this leads to a nice simplification, there is no *requirement* that these choices be made. We can use *any* transitions as long as the ratio $U_m/T_m$ is tractable.

We propose to use the "uncorrected" versions of the transitions $T_m$ and $U_m$ used by Hamiltonian AIS, obtained by dropping the accept-reject steps. To get an expression for the uncorrected $U_m$ we first derive the reversal $U_m$ used by Hamiltonian AIS (Alg. 2). These uncorrected transitions are no longer reversible with respect to the bridging densities $\pi_m(z, \rho)$, and thus we cannot use the simplification used by AIS to get eq. 8. Despite this, we show that the ratio $U_m/T_m$ for the uncorrected transitions can still be easily computed (Thm. 2). This produces a differentiable estimator, meaning the parameters can be tuned by stochastic gradient methods designed to maximize the ELBO.

We start by deriving the process that defines the transition $U_m$ used by Hamiltonian AIS. This is shown in Alg. 2. It can be observed that $U_m$ follows the same three steps of $T_m$ (resample momentum, Hamiltonian simulation with accept-reject, momentum negation), but in reverse order.

---
**Algorithm 2** Corrected $U_m(z_m, \rho_m | z_{m+1}, \rho_{m+1})$

1. Set $(z_m''', \rho_m''') \leftarrow (z_{m+1}, -\rho_{m+1})$.
2. Simulate Hamiltonian dynamics as $(z_m'', \rho_m'') \leftarrow \mathcal{T}_m(z_m''', \rho_m''')$.
   Calculate an acceptance probability $\alpha = \min(1, \bar{\pi}_m(z_m'', \rho_m'')/\bar{\pi}_m(z_m''', \rho_m'''))$.
   With probability $\alpha$, set $(z_m', \rho_m') \leftarrow (z_m'', \rho_m'')$. Otherwise, set $(z_m', \rho_m') \leftarrow (z_m''', \rho_m''')$.
3. Sample $\rho_m$ from $s_{\text{rev}}(\rho_m | \rho_m')$, the reversal of $s(\rho_m' | \rho_m)$ with respect to $S(\rho_m)$. Set $z_m \leftarrow z_m'$.
**return** $(z_m, \rho_m)$

---

**Lemma 1.** *The corrected $U_m$ (Alg. 2) is the reversal of the corrected $T_m$ (Alg. 1) with respect to $\pi_m$.*

*(Proof Sketch).* First, we claim the general result that if $T_1$, $T_2$ and $T_3$ have reversals $U_1$, $U_2$ and $U_3$, respectively, then the composition $T = T_1 \circ T_2 \circ T_3$ has reversal $U = U_3 \circ U_2 \circ U_1$ (all reversals with respect to same density $\pi$). Then, we apply this to the corrected $T_m$ and $U_m$: $T_m$ is the composition of three steps that hold $\pi_m$ invariant. Thus, its reversal $U_m$ is given by the composition of the reversals of those steps, applied in reversed order. A full proof is in Appendix F. $\qquad\square$

We now define the "uncorrected" transitions used by UHA, shown in Algs. 3 and 4. These are just the transitions used by Hamiltonian AIS but without the accept-reject steps. (If Hamiltonian dynamics are simulated exactly, the acceptance rate is one and the uncorrected and corrected transitions are equivalent.) We emphasize that, for the "uncorrected" transitions, $T_m$ does not exactly hold $\pi_m$ invariant and $U_m$ is not the reversal of $T_m$. Thus, their ratio does not give a simple expression in terms of $\bar{\pi}_m$ as in eq. 8. Nevertheless, the following result shows that their ratio has a simple form.

**Theorem 2.** *Let $T_m$ and $U_m$ be the uncorrected transitions defined in Algs. 3 and 4, and let the dynamics simulator $\mathcal{T}_m(z, \rho)$ be volume preserving and self inverting. Then,*

$$\frac{U_m(z_m, \rho_m | z_{m+1}, \rho_{m+1})}{T_m(z_{m+1}, \rho_{m+1} | z_m, \rho_m)} = \frac{S(\rho_m)}{S(\rho_m')}, \tag{10}$$

---

**Algorithm 3** Uncorrected $T_m(z_{m+1}, \rho_{m+1} | z_m, \rho_m)$

---

1. Sample $\rho'_m$ from some $s(\rho'_m | \rho_m)$ that leaves $S(\rho)$ invariant. Set $z'_m \leftarrow z_m$.
2. Simulate Hamiltonian dynamics as $(z''_m, \rho''_m) \leftarrow \mathcal{T}_m(z'_m, \rho'_m)$.
3. Reverse the momentum as $(z_{m+1}, \rho_{m+1}) \leftarrow (z''_m, -\rho''_m)$.

**return** $(z_{m+1}, \rho_{m+1})$

---

**Algorithm 4** Uncorrected $U_m(z_m, \rho_m | z_{m+1}, \rho_{m+1})$

---

1. Set $(z''_m, \rho''_m) \leftarrow (z_{m+1}, -\rho_{m+1})$.
2. Simulate Hamiltonian dynamics as $(z'_m, \rho'_m) \leftarrow \mathcal{T}_m(z''_m, \rho''_m)$.
3. Sample $\rho_m$ from $s_{\text{rev}}(\rho_m | \rho'_m)$, the reversal of $s(\rho'_m | \rho_m)$ with respect to $S(\rho_m)$. Set $z_m \leftarrow z'_m$.

**return** $(z_m, \rho_m)$

---

*where $\rho'_m$ is the second component of $\mathcal{T}_m(z_{m+1}, -\rho_{m+1})$. (That is, $\rho'_m$ from Algs. 3 and 4.)*

*(Proof Sketch.)* We consider variants of Algs. 3 and 4 in which each time $z$ is assigned we add Gaussian noise with some variance $aI$. We then derive the densities for $T_m$ and $U_m$ using the rule for transformation of densities under invertible mappings, using that $\mathcal{T}_m$ is self-inverting and volume preserving. Taking the ratio gives eq. 10. Since this is true for arbitrary $a$, we take the stated result as the limit as $a \to 0$. A full proof is in Appendix G. $\qquad\square$

As an immediately corollary of eq. 9 and Theorem 2 we get that for UHA

$$\frac{\bar{p}(z_{1:M}, \rho_{1:M})}{q(z_{1:M}, \rho_{1:M})} = \frac{\bar{p}(z_M)}{q(z_1)} \prod_{m=1}^{M-1} \frac{S(\rho_{m+1})}{S(\rho'_m)}. \tag{11}$$

This ratio can be used to get an expression for the lower bound $\text{ELBO}(q(z_{1:M}, \rho_{1:M}), \bar{p}(z_{1:M}, \rho_{1:M}))$ obtained with UHA. As mentioned in Section 2, the parameters of the augmented distributions are tuned to maximize the ELBO, equivalent to minimizing the KL-divergence from $q$ to $\bar{p}$. While computing this ELBO exactly is typically intractable, an unbiased estimate can be obtained using a sample from $q(z_{1:M}, \rho_{1:M})$ as shown in Alg. 5. If sampling is done using reparameterization, then unbiased reparameterization gradients may be used together with stochastic optimization algorithms to optimize the lower bound. In contrast, the variational lower bound obtained with Hamiltonian AIS (see Alg. 6 in Appendix A) does not allow the computation of unbiased reparameterization gradients.

---

**Algorithm 5** Generating the (differentiable) uncorrected Hamiltonian annealing variational bound.

---

Sample $z_1 \sim q$ and $\rho_1 \sim S$.
Initialize estimator as $\mathcal{L} \leftarrow -\log q(z_1)$.
**for** $m = 1, 2, \cdots, M-1$ **do**
    Run uncorrected $T_m$ (Alg. 3) on input $(z_m, \rho_m)$, storing $\rho'_m$ and the output $(z_{m+1}, \rho_{m+1})$.
    Update estimator as $\mathcal{L} \leftarrow \mathcal{L} + \log\left(S(\rho_{m+1})/S(\rho'_m)\right)$.
Update estimator as $\mathcal{L} \leftarrow \mathcal{L} + \log \bar{p}(z_M)$.
**return** $R$

---

### 3.1 Algorithm Details

**Simulation of dynamics.** We use the leapfrog operator with step-size $\epsilon$ to simulate Hamiltonian dynamics. This has unit Jacobian and satisfies $\mathcal{T}_m = \mathcal{T}_m^{-1}$ (if the momentum is negated after the simulation), which are the properties required for eq. 11 to be correct (see Theorem 2).

**Momentum distribution and resampling.** We set the momentum distribution $S(\rho) = \mathcal{N}(\rho | 0, \Sigma)$ to be a Gaussian with mean zero and covariance $\Sigma$. The resampling distribution $s(\rho' | \rho)$ must hold this distribution invariant. As is common we use $s(\rho' | \rho) = \mathcal{N}(\rho' | \eta \rho, (1 - \eta^2)\Sigma)$, where $\eta \in [0, 1)$ is the damping coefficient. If $\eta = 0$, the momentum is completely replaced with a new sample from $S$ in each iteration (used by HMC and Langevin dynamics [29, 42]). For larger $\eta$, the momentum becomes correlated between iterations, which may help suppress random walk behavior and encourage faster mixing [9] (used by the underdamped variants of HMC and Langevin dynamics [29]).

**Bridging densities.** We set $\bar{\pi}_m(z, \rho) = q(z, \rho)^{1-\beta_m} \bar{p}(z, \rho)^{\beta_m}$, where $\beta_m \in [0, 1]$ and $\beta_m < \beta_{m+1}$.

**Computing gradients.** We set the initial distribution $q(z_1)$ to be a Gaussian, and perform all sampling operations in Alg. 5 using reparameterization [24, 31, 39]. Thus, the whole procedure is differentiable and reparameterization-based gradients may be used to tune parameters by maximizing the ELBO. These parameters include the initial distribution $q(z_1)$, the covariance $\Sigma$ of the momentum distribution, the step-size $\epsilon$ of the integrator, the damping coefficient $\eta$ of the momentum resampling distribution, and the parameters of the bridging densities (including $\beta$), among others. As observed in Section 5.2.1 tuning all of these parameters may lead to considerable performance improvements.

## 4 Related Work

UHA and slight variations have been proposed in concurrent work by Thin et al. [38], who use uncorrected Langevin dynamics together with the uncorrected reversal to build variational lower bounds, and by Zhang et al. [46], who proposed UHA with under-damped Langevin dynamics together with a convergence analysis for linear regression models.

There are three other lines of work that produce differentiable variational bounds integrating Monte Carlo methods. One is Hamiltonian VI (HVI) [34, 43]. It uses eq. 9 to build a lower bound on $\log Z$, with $T_m$ set to an uncorrected Hamiltonian transition (like UHA but without bridging densities) and $U_m$ set to conditional Gaussians parameterized by learnable functions. Typically, a single transition is used, and the parameters of the transitions are learned by maximizing the resulting ELBO.[1]

A second method is given by Hamiltonian VAE (HVAE) [7], based on Hamiltonian Importance sampling [28]. They augment the variational distribution with momentum variables, and use the leapfrog integrator to simulate Hamiltonian dynamics (a deterministic invertible transformation with unit Jacobian) with a tempering scheme as a target-informed flow [30, 37].

The third method is Importance Weighting (IW) [6, 13, 15]. Here, the idea is that $\text{ELBO}(q(z), \bar{p}(z)) \leq \mathbb{E} \log \frac{1}{K} \sum_k \bar{p}(z_k)/q(z_k)$, and that the latter bound can be optimized, rather than the traditional ELBO. More generally, other Monte-Carlo estimators can be used [16].

Some work defines novel contrastive-divergence-like objectives in terms of the final iteration of an MCMC chain [32, 26]. These do not provide an ELBO-like variational bound. While in some cases the initial distribution can be optimized to minimize the objective [32], gradients do not flow through the MCMC chains, meaning MCMC parameters cannot be optimized by gradient methods.

For latent variable models, Hoffman [19] suggested to run a few MCMC steps after sampling from the variational distribution before computing gradients with respect to the model parameters, which is expected to "debias" the gradient estimator to be closer to the true likelihood gradient. The variational distribution is simultaneously learned to optimize a standard ELBO. (AIS can also be used [14].)

## 5 Experiments and Results

This section presents results using UHA for Bayesian inference problems on several models of varying dimensionality and for VAE training. We compare against Hamiltonian AIS, IW, HVI and HVAE. We report the performance of each method for different values of $K$, the number of likelihood evaluations required to build the lower bound (e.g. number of samples used for IW, number of bridging densities plus one for UHA). Note that, for a fixed $K$, all methods have the same oracle complexity (i.e. number of target/target's gradient evaluation), and that for $K = 1$ they all reduce to plain VI.

For UHA and Hamiltonian AIS we use under-damped Langevin dynamics, that is, we perform just one leapfrog step per transition and partially resample momentum. We implement all algorithms using Jax [5].

---

[1]The formulation of HVI allows the use of more than one transition. However, this leads to an increased number of reverse models that must be learned, and thus not typically used in practice. Indeed, experiments by Salimans et al. [34] use only one HMC step while varying the number of leapfrog integration steps, and results from Wolf et al. [43] show that increasing the number of transitions may actually yield worse bounds (they conjecture that this is due to the difficulty of learning inverse dynamics.).

Table 1: **Our method (UHA) yields better bounds than importance weighting (IW) for moderate or high dimensions.** ELBO achieved by different methods when using a Student-t target distribution of varying dimensionality, higher is better. Since the target is normalized, a perfect inference algorithm would achieve the true value of $\log Z = 0$.

| Target dimension | Plain VI $K = 1$ | UHA | | | | IW | |
|---|---|---|---|---|---|---|---|
| | | $K = 4$ | $K = 16$ | $K = 64$ | $K = 128$ | $K = 128$ | $K = 1024$ |
| 20 | $-0.82$ | $-0.55$ | $-0.36$ | $-0.19$ | $-0.14$ | $-0.14$ | $-0.088$ |
| 200 | $-8.1$ | $-5.5$ | $-3.5$ | $-1.9$ | $-1.4$ | $-3.7$ | $-2.9$ |
| 500 | $-20.5$ | $-13.9$ | $-9.0$ | $-5.2$ | $-3.8$ | $-12.0$ | $-10.4$ |

### 5.1 Toy example

This section compares results obtained with UHA and IW when the target is set to a factorized Student-t with mean zero, scale one, and three degrees of freedom. We tested three different dimensionalities: 20, 200 and 500. In all cases we have $\log Z = 0$, so we can exactly analyze the tightness of the bounds obtained by the methods. We set the initial approximation to be a mean-field Gaussian, and optimize the objective using Adam [23] with a step-size of 0.001 for 5000 steps. For UHA we tune the initial approximation $q(z)$, the integrator's step-size $\epsilon$ and the damping coefficient $\eta$.

We ran UHA for $K \in \{4, 16, 64, 128\}$ and IW for $K \in \{128, 1024\}$. Table 1 shows the results for the three dimensionalities considered. It can be observed that UHA performs significantly better than IW as the dimensionality increases; for the target with dimension 500, UHA with $K = 16$ yields better bounds than IW with $K = 1024$. On the other hand, the methods perform similarly for the low dimensional target. Finally, in this case both methods have similar time costs. For instance, for $K = 128$ UHA takes 14.2 seconds to optimize and IW takes 13.9.

### 5.2 Inference tasks

This section shows results using UHA for Bayesian inference tasks. For this set of experiments, for UHA we tune the initial distribution $q(z)$, the integrator's step-size $\epsilon$ and the damping coefficient $\eta$. We include detailed results tuning more parameters in Section 5.2.1.

**Models.** We consider four models: *Brownian motion* ($d = 32$), which models a Brownian Motion process with a Gaussian observation model; *Convection Lorenz bridge* ($d = 90$), which models a nonlinear dynamical system for atmospheric convection; and *Logistic regression* with the a1a ($d = 120$) and madelon ($d = 500$) datasets. The first two obtained from the "Inference gym" [36].

**Baselines.** We compare UHA against IW, HVAE, a simple variant of HVI, and Hamiltonian AIS (HAIS). For all methods which rely on HMC (i.e. all except IW) we use a singe integration step-size $\epsilon$ common to all dimensions and fix the momentum distribution to a standard Gaussian. For HVI we learn the initial distribution $q(z)$, integration step-size $\epsilon$ and the reverse dynamics $U_m$ (set to a factorized Gaussian with mean and variance given by affine functions), and for HVAE we learn $q(z)$, $\epsilon$ and the tempering scheme (we use the quadratic scheme parameterized by a single parameter).

**Training details.** We set $q(z)$ to be a mean-field Gaussian initialized to a maximizer of the ELBO, and tune the parameters of each method by running Adam for 5000 steps. We repeat all simulations for different step-sizes in $\{10^{-3}, 10^{-4}, 10^{-5}\}$, and select the best one for each method. Since Hamiltonian AIS' parameters cannot be tuned by gradient descent, we find a good pair $(\epsilon, \eta)$ by grid search. We consider $\eta \in \{0.5, 0.9, 0.99\}$ and three values of $\epsilon$ that correspond to three different rejection rates: 0.05, 0.25 and 0.5. We tested all 9 possible combinations and selected the best one.

Results are shown in Fig. 1. Our method yields better lower bounds than all other competing approaches for all models considered, and that increasing the number of bridging densities consistently leads to better results. The next best performing method is Hamiltonian AIS. IW also shows a good performance for the lower dimensional model *Brownian motion*. However, for models of higher dimensionality IW leads to bounds that are several nats worse than the ones achieved by UHA. Finally, HVI and HVAE yield bounds that are much worse than those achieved by the other three methods, and do not appear to improve consistently for larger $K$. For HVAE, these results are consistent with

the ones in the original paper [7, §4], in that higher $K$ may sometimes hurt performance. For HVI, we believe this is related to the use of just one HMC step and suboptimal inverse dynamics.

Optimization times for Plain VI, IW and UHA (the latter two with $K = 32$) are $2.4, 3.4$ and $4.4$ seconds for the *Brownian motion* dataset, $2.5, 6.8$ and $6.9$ seconds for *Lorenz convection*, $2.8, 8.3$ and $19.9$ seconds for *Logistic regression (A1A)*, and $4.6, 16.6$ and $121.2$ seconds for *Logistic regression (Madelon)*. While IW and UHA have the same oracle complexity for the same $K$, we see that the difference between their time cost depends on the specific model under consideration. All other methods that use HMC have essentially the same time cost as UHA.

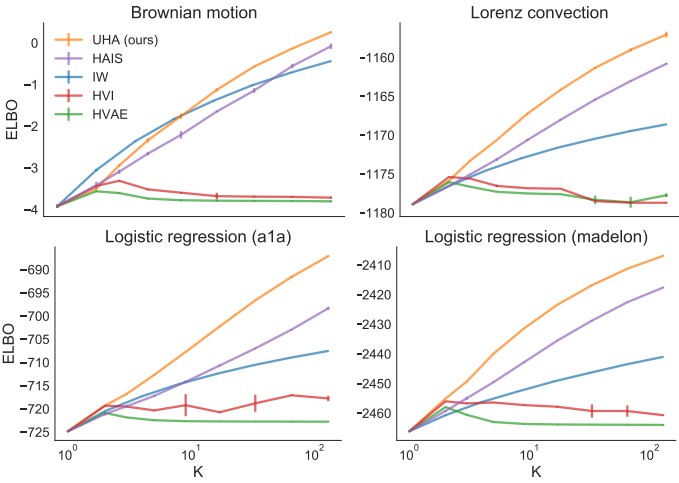

Figure 1: **Our method achieves much better bounds than other competing methods.** $K$ represents the number of likelihood evaluations to build the lower bound required by each method. The leftmost point of all lines coincide because, for $K = 1$, all methods reduce to plain VI. Vertical bars indicate one standard deviation obtained by running simulations with four different random seeds.

### 5.2.1 Tuning More Parameters with UHA

A basic version of UHA involves fitting a variational distribution using plain VI, and then tuning the integration step-size $\epsilon$ and the damping coefficient $\eta$. However, more parameters could be tuned:

- Moment distribution cov $\Sigma$: We propose to learn a diagonal matrix instead of using the identity.
- Bridging densities' coefficients $\beta_m$: Typically $\beta_m = m/M$. We propose to learn the sequence $\beta$, with the restrictions $\beta_0 = 0, \beta_M = 1, \beta_m < \beta_{m+1}$ and $\beta_m \in [0, 1]$.
- Initial distribution $q(z)$: Instead of fixing $q(z)$ to be a maximizer of the typical ELBO, we propose to learn it to maximize the augmented ELBO obtained using UHA.
- Integrator's step-size $\epsilon$: Instead of learning a unique step-size $\epsilon$, we propose to learn a step-size that is a function of $\beta$, i.e. $\epsilon(\beta)$. In our experiments we use an affine function.
- Bridging densities parameters $\psi$: Instead of setting the $m$-th bridging density to be $q^{1-\beta_m} p^{\beta_m}$, we propose to set it to $q_{\psi(\beta_m)}^{1-\beta_m} p^{\beta_m}$, where $q_{\psi(\beta_m)}$ is a mean-field Gaussian with a mean and diagonal covariance specified as affine functions of $\beta$.

We consider the four models described previously and compare three methods: UHA tuning all parameters described above, UHA tuning only the pair $(\epsilon, \eta)$, and Hamiltonian AIS with parameters $(\epsilon, \eta)$ obtained by grid-search. We perform the comparison for $K$ ranging from 2 to 512. (For $K \geq 64$ we tune the UHA's parameters using $K = 64$ and extrapolate them as explained in Appendix D.)

Results are shown in Fig. 2. It can be observed that tuning all parameters with UHA leads to significantly better lower bounds than those obtained by Hamiltonian AIS (or UHA tuning only $\epsilon$ and $\eta$). Indeed, for the Logistic regression models, UHA tuning all parameters for $K = 64$ leads to results comparable to the ones obtained by Hamiltonian AIS with $K = 512$.

To verify what parameters lead to larger performance improvements, we tested UHA with $K = 64$ tuning different subsets of $\{\epsilon, \eta, \Sigma, \beta, q(z), \epsilon(\beta), \psi(\beta)\}$. Fig. 3 shows the results. It can be observed

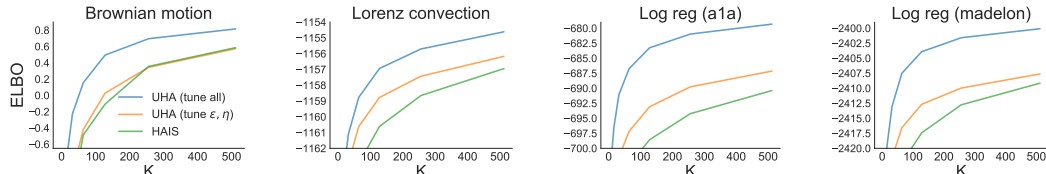

Figure 2: **UHA tuning all parameters leads to better performance than other methods.**

that tuning the bridging parameters $\beta$ and the initial approximation $q(z)$ leads to the largest gains in performance, and that tuning all parameters always outperforms tuning smaller subsets of parameters. We show a more thorough analysis, including more subsets and values of $K$ in Appendix B.

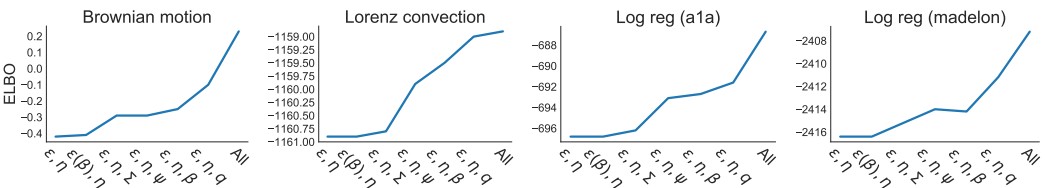

Figure 3: **Tuning all parameters leads to better results than tuning subsets of them. Largest gains are obtained by tuning bridging coefficients $\beta$ and initial distribution $q$.** ELBO achieved as a function of parameters tuned (x-axis), for $K = 64$. The subsets are ordered in terms of increasing performance (same ordering is used for all four models). Parameters are step-size $\epsilon$, damping coefficient $\eta$, moment covariance $\Sigma$, bridging densities parameters $\beta$ and $\psi$, initial distribution $q$.

Finally, Appendix E shows results comparing UHA (tuning several parameters) against HMC, mean field VI and IW in terms of the approximation accuracy achieved on a logistic regression model with a fixed computational budget.

## 5.3   VAE training

Our method can be used to train latent variable models, such as Variational Auto-encoders (VAE) [24, 31]. In this case the initial approximation $q(z|x)$ and the model $p(x, z)$ are parameterized by two neural networks (encoder and decoder), whose parameters are trained by maximizing the ELBO. UHA can be used to train VAEs by augmenting these two distributions as described in Section 3.

**Datasets.** We use three datasets: mnist [25] (numbers 1-9), emnist-letters [11] (letters A-Z), and kmnist [10] (cursive Kuzushiji). All consist on greyscale images of $28 \times 28$ pixels. In all cases we use stochastic binarization [33] and a training set of 50000 samples, a validation set of 10000 samples, and a test set of 10000 samples. All datasets are available in tensorflow-datasets [1].

**Baselines.** We compare against Importance Weighted Auto-encoders [6] and plain VAE training [24].

**Architecture details.** We set $q(z|x)$ to a diagonal Gaussian, $p(z)$ to a standard Normal, and $p(x|z)$ to a Bernoulli. We consider two architectures for the encoder and decoder: (1) Feed forward networks with one hidden layer of size 450 and Relu non-linearities, with a latent space dimensionality of 64; (2) Architecture used by Burda et al. [6], feed forward networks with two hidden layers of size 200 with tanh non-linearities, with a latent space dimensionality of 50.

**Training details.** In all cases the encoder and decoder are initialized to parameters that maximize the ELBO. For IW we tune the encoder and decoder parameters (using the doubly-reparameterized estimator [40]), and for UHA we tune the integration step-size $\epsilon$, damping coefficient $\eta$, bridging parameters $\beta$, momentum covariance $\Sigma$ (diagonal), and the decoder parameters. Following Caterini et al. [7] we constrain $\epsilon \in (0, 0.05)$ to avoid unstable behavior of the leapfrog discretization. We use Adam with a step-size of $10^{-4}$ to train for 100 epochs and use the validation set for early stopping. We repeated all simulations for three different random seeds. In all cases the standard deviation of the results was less than 0.1 nats (not shown in tables).

All methods achieved better results using the architecture with one hidden layer. These results are shown in Tables 2 and 3. The first one shows the ELBO on the test set achieved for different values of $K$, and the second one the log-likelihood on the test set estimated with AIS [44]. It can be observed that UHA leads to higher ELBOs, higher log-likelihoods, and smaller variational gaps (difference between ELBO and log-likelihood) than IW for all datasets, with the difference between both methods' performance increasing for increasing $K$. Notably, for $K = 64$, the variational gap for UHA becomes quite small, ranging from $0.8$ to $1.4$ nats depending on the dataset.

Results for the architecture from Burda et al. [6] (two hidden layers) are shown in Tables 4 and 5 (Appendix C). Again, we observe that UHA consistently leads to higher ELBOs and the best test log-likelihood was consistently achieved by UHA with $K = 64$. However, for smaller $K$, IW sometimes had better log-likelihoods than UHA (despite worse ELBOs).

Table 2: ELBO on the test set (higher is better). For $K = 1$ both methods reduce to plain VI.

|  |  | $K = 1$ | $K = 8$ | $K = 16$ | $K = 32$ | $K = 64$ |
|---|---|---|---|---|---|---|
| mnist | UHA | $-93.4$ | $-89.8$ | $-88.8$ | $-88.1$ | $-87.6$ |
|  | IW | $-93.4$ | $-90.5$ | $-89.9$ | $-89.4$ | $-89.0$ |
| letters | UHA | $-137.9$ | $-133.5$ | $-132.3$ | $-131.5$ | $-130.9$ |
|  | IW | $-137.9$ | $-134.6$ | $-133.9$ | $-133.2$ | $-132.7$ |
| kmnist | UHA | $-184.2$ | $-176.6$ | $-174.6$ | $-173.2$ | $-171.6$ |
|  | IW | $-184.2$ | $-179.7$ | $-178.7$ | $-177.8$ | $-177.0$ |

Table 3: Log-likelihood on the test set (higher is better). This is estimated using AIS with underdamped HMC using 2000 bridging densities, 1 HMC iteration with 16 leapfrog steps per bridging density, integration step-size $\epsilon = 0.06$, and damping coefficient $\eta = 0.8$.

|  |  | $K = 1$ | $K = 8$ | $K = 16$ | $K = 32$ | $K = 64$ |
|---|---|---|---|---|---|---|
| mnist | UHA | $-88.5$ | $-87.5$ | $-87.2$ | $-87.0$ | $-86.9$ |
|  | IW | $-88.5$ | $-87.6$ | $-87.5$ | $-87.3$ | $-87.2$ |
| letters | UHA | $-131.9$ | $-130.7$ | $-130.3$ | $-130.1$ | $-129.9$ |
|  | IW | $-131.9$ | $-130.9$ | $-130.7$ | $-130.6$ | $-130.4$ |
| kmnist | UHA | $-174.3$ | $-172.2$ | $-171.6$ | $-171.2$ | $-170.2$ |
|  | IW | $-174.3$ | $-173.0$ | $-172.6$ | $-172.4$ | $-172.2$ |

## 6   Discussion

Since UHA yields a differentiable lower bound, one could tune other parameters not considered in this work. For instance, a different momentum distribution per bridging density could be used, that is, $\bar{\pi}_m(z, \rho) = \bar{\pi}_m(z) S_m(\rho)$. We believe additions such as this may yield further gains. Also, our method can be used to get tight and differentiable upper bounds on $\log Z$ using the reversed AIS procedure described by Grosse et al. [18].

Finally, removing accept-reject steps might sometimes lead to instabilities during optimization if the step-size $\epsilon$ becomes large. We observed this effect when training VAEs on some datasets for the larger values of $K$. We solved this by constraining the range of $\epsilon$ (previously done by Caterini et al. [7]). While this simple solution works well, we believe that other approaches (e.g. regularization, automatic adaptation) could work even better. We leave the study of such alternatives for future work.

## Acknowledgments and Disclosure of Funding

This material is based upon work supported in part by the National Science Foundation under Grant No. 1908577.

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
