## A  Generating the Hamiltonian AIS bound

---

**Algorithm 6** Generating the (non-differentiable) Hamiltonian AIS variational bound.

---

Sample $z_1 \sim q$ and $\rho_1 \sim S$.
Initialize estimator as $\mathcal{L} \leftarrow -\log q(z_1, \rho_1)$.
**for** $m = 1, 2, \cdots, M - 1$ **do**
    Run corrected $T_m$ (Alg. 1) on input $(z_m, \rho_m)$, storing the output $(z_{m+1}, \rho_{m+1})$.
    Update estimator as $\mathcal{L} \leftarrow \mathcal{L} + \log \left( \bar{\pi}_m(z_m, \rho_m) / \bar{\pi}_m(z_{m+1}, \rho_{m+1}) \right)$.
Update estimator as $\mathcal{L} \leftarrow \mathcal{L} + \log \bar{p}(z_M, \rho_M)$.
**return** $\mathcal{L}$

---

## B  More results tuning more subsets of parameters for UHA

We tested UHA tuning different subsets of $\{\epsilon, \eta, \Sigma, \beta, q(z), \epsilon(\beta), \psi(\beta)\}$. Fig. 4 shows the results. The first row shows the results obtained by tuning the pair $(\epsilon, \eta)$ and each other parameter individually for different values of $K$, and the second row shows the results obtained by tuning increasingly more parameters. It can be observed that tuning $\beta$ and $q(z)$ lead to the largest gains in performance.

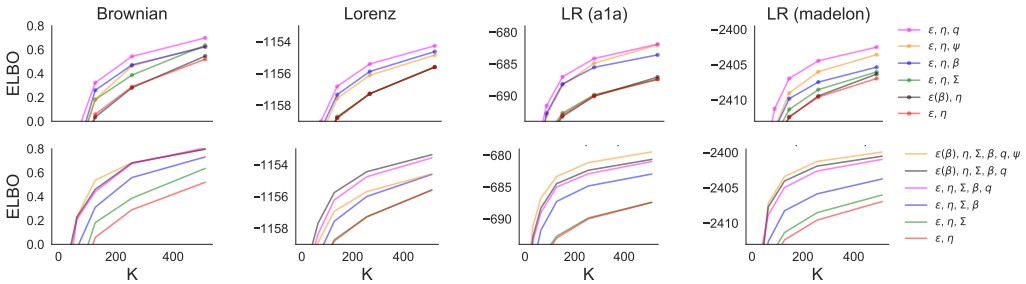

Figure 4: **Tuning more parameters leads to significantly better results.** Legends indicate what parameters are being trained. Parameters are step-size $\epsilon$, damping coefficient $\eta$, moment covariance $\Sigma$, bridging densities parameters $\beta$ and $\psi$, initial distribution $q$. $\epsilon(\beta)$ indicates we are learning the step-size as an affine function of $\beta$.

## C  Results using architecture from Burda et al. [6]

In this section we show the results achieved for VAE training using the architecture from Burda et al. [6] (with 1 stochastic layer). In this case the encoder and decoder consist on feed forward neural networks with two hidden layers of size 200 with *Tanh* non-linearity, and latent space dimensionality of 50. All training details are the same, but with the constraint $\epsilon \in (0, 0.04)$. Tables 4 and 5 show the results.

Table 4: ELBO on the test set (higher is better). For $K = 1$ both methods reduce to plain VI.

|         |     | $K = 1$  | $K = 8$   | $K = 16$  | $K = 32$  | $K = 64$  |
|---------|-----|----------|-----------|-----------|-----------|-----------|
| mnist   | UHA | $-92.4$  | $-89.2$   | $-88.5$   | $-88.1$   | $-87.1$   |
|         | IW  | $-92.4$  | $-89.9$   | $-89.3$   | $-88.8$   | $-88.5$   |
| letters | UHA | $-139.0$ | $-134.3$  | $-133.3$  | $-132.6$  | $-131.2$  |
|         | IW  | $-139.0$ | $-135.5$  | $-134.7$  | $-134.0$  | $-133.4$  |
| kmnist  | UHA | $-197.5$ | $-189.5$  | $-188.1$  | $-187.1$  | $-180.3$  |
|         | IW  | $-197.5$ | $-191.8$  | $-190.2$  | $-188.8$  | $-187.6$  |

Table 5: Log-likelihood on the test set (higher is better). This is estimated using AIS with underdamped HMC using 2000 bridging densities, 1 HMC iteration with 16 leapfrog steps per bridging density, integration step-size $\epsilon = 0.05$, and damping coefficient $\eta = 0.8$.

|  |  | $K = 1$ | $K = 8$ | $K = 16$ | $K = 32$ | $K = 64$ |
|---|---|---|---|---|---|---|
| mnist | UHA | $-88.3$ | $-87.6$ | $-87.4$ | $-87.3$ | $-86.3$ |
|  | IW | $-88.3$ | $-87.3$ | $-87.0$ | $-86.8$ | $-86.6$ |
| letters | UHA | $-133.0$ | $-131.8$ | $-131.4$ | $-131.2$ | $-129.9$ |
|  | IW | $-133.0$ | $-131.6$ | $-131.2$ | $-130.9$ | $-130.6$ |
| kmnist | UHA | $-188.3$ | $-186.3$ | $-185.8$ | $-185.3$ | $-177.4$ |
|  | IW | $-188.3$ | $-184.4$ | $-183.2$ | $-182.1$ | $-181.2$ |

## D  Extrapolating optimal parameters for UHA

Some results in Section 5.2.1 (and Appendix B) use a number of bridging densities $K$ up to 512. As mentioned previously, for those simulations, if $K_1 \geq 64$ bridging densities were used, we optimized the parameters for $K_2 = 64$ and extrapolate the parameters to work with $K_1$. We now explain this procedure.

From the parameters considered, $\{\epsilon, \eta, \Sigma, \beta, q(z), \epsilon(\beta), \psi(\beta)\}$, the only ones that need to be "extrapolated" are the step-size $\epsilon$ and the bridging parameters $\beta$. All other parameters are tuned for $K_2 = 64$ bridging densities and the values obtained are directly used with $K_1$ bridging densities.

For $\beta$ we use a simple interpolation. Define $f(x)$ to be the piecewise linear function (with $K_2 = 64$ "pieces") that satisfies $f(x_k) = \beta_k$, for $x_k = k/K_2$ and $k = 0, \cdots, K_2$ (this is a bijection from $[0, 1]$ to $[0, 1]$). Then, when using $K_1$, we simply define $\beta_k = f(x_k)$, where $x_k = k/K_1$ and $k = 0, \cdots, K_1$.

For $\epsilon$, we use the transformation $\epsilon_{K_1} = \epsilon_{K_2} \frac{\log K_2}{\log K_1}$. While other transformations could be used (e.g. without the log), we observed this to work best in practice. (In fact, we obtained this rule by analyzing the dependence of the optimal $\epsilon$ on $K$ for several tasks and values of $K$.)

## E  Approximation accuracy

We study the accuracy of the approximation provided by UHA by analyzing the posterior moment errors: We estimate the mean and covariance of the target distribution using UHA and compute the mean absolute error of these estimates. (We get the ground truth values using approximate samples obtained running NUTS [20] for 500000 steps.) We consider a logistic regression model with the *sonar* dataset ($d = 61$), and compare against mean field VI, IW, and HMC. We give each method the same computational budget $B$, measured as the total number of model evaluations (or gradient), and perform simulations for $B \in \{10^5, 5 \times 10^5, 10^6\}$.

For HMC, we use half of the budget for the warm-up phase and half to draw samples. For mean field VI we use the whole budget for optimization, and use the final mean and variance parameters for the approximation. For UHA and IW we train using $K = 32$ for 3000 steps, and use the remaining budget of model evaluations to draw samples (used to estimate posterior moments) using $K = 256$.[2] For UHA we tune the step-size $\epsilon$, the damping coefficient $\eta$, the momentum distribution covariance (diagonal), the bridging densities coefficients $\beta$, and the parameters of the initial distribution $q(z)$.

Fig. 5 shows the results for the posterior covariance. We do not include the results for the posterior mean because all methods perform similarly. It can be observed that HMC achieves the lowest error, followed by UHA. Both mean field VI and IW yield significantly worse results.

## F  Proof of Lemma 1

We begin with the following result.

---

[2]For UHA we use the extrapolation explained in Appendix D

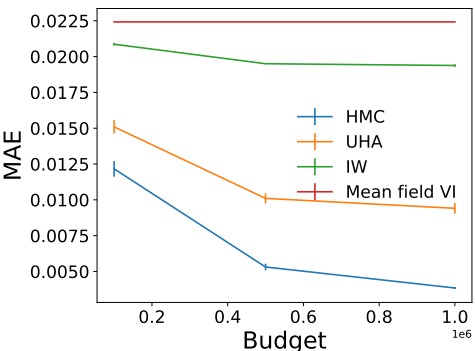

Figure 5: Mean absolute error for posterior covariance approximation. Standard errors computed by repeating the simulations using five different random seeds.

**Lemma 3.** *Let $T_1$, $T_2$ and $T_3$ be three transitions that leave some distribution $\pi$ invariant and satisfy $T_i(z'|z)\pi(z) = U_i(z|z')\pi(z')$ (i.e. $U_i$ is the reversal of $T_i$ with respect to $\pi$). Then the reversal of $T$ with respect to $\pi$ is given by $U = U_3 \circ U_2 \circ U_1$.*

*Proof.*

$$T(z'|z)\pi(z) = \int T_3(z'|z_2)T_2(z_2|z_1)T_1(z_1|z)\pi(z)\,dz_1\,dz_2 \tag{12}$$

$$= \int T_3(z'|z_2)T_2(z_2|z_1)\pi(z_1)U_1(z|z_1)\,dz_1\,dz_2 \tag{13}$$

$$= \int T_3(z'|z_2)\pi(z_2)U_2(z_1|z_2)U_1(z|z_1)\,dz_1\,dz_2 \tag{14}$$

$$= \pi(z')\int U_3(z_2|z')U_2(z_1|z_2)U_1(z|z_1)\,dz_1\,dz_2 \tag{15}$$

$$= \pi(z')\int U_1(z|z_1)U_2(z_1|z_2)U_3(z_2|z')\,dz_1\,dz_2 \tag{16}$$

$$= \pi(z')U(z|z'). \tag{17}$$

$\square$

The rest of the proof is straightforward. Let the three steps from the corrected version of $T_m$ (Alg. 1) be denoted $T_m^1$, $T_m^2$ and $T_m^3$. The latter two (Hamiltonian simulation with accept-reject step and momentum negation) satisfy detailed balance with respect to $\pi_m(z, \rho)$ [28, §3.2]. Thus, for these two, $U_m^i$ is defined by the same process as $T_m^i$. For $T_m^1$ (momentum resampling), its reversal is given by the reversal of $s(\rho'|\rho)$ with respect to $S(\rho)$. We call this $s_{\text{rev}}(\rho|\rho')$, and it satisfies

$$s_{\text{rev}}(\rho|\rho') = s(\rho'|\rho)\frac{S(\rho)}{S(\rho')}. \tag{18}$$

## G  Proof of Theorem 2

To deal with delta functions, whenever the transition states [Set $z' \leftarrow z$], we use $z' \sim \mathcal{N}(z, a)$, and take the limit $a \to 0$. We use $g_a(z)$ to denote the density of a Gaussian with mean zero and variance $a$ evaluated at $z$, and $\gamma(z, \rho) = (z, -\rho)$ (operator that negates momentum).

We first compute $T_m(z_{m+1}, \rho_{m+1}|z_m, \rho_m)$. We have that $\rho'_m \sim s(\cdot|\rho_m)$ and $z'_m \sim \mathcal{N}(z_m, a)$. Thus,

$$T'_m(z'_m, \rho'_m|z_m, \rho_m) = s(\rho'_m|\rho_m)g_a(z'_m - z_m). \tag{19}$$

Also, we have $(z_{m+1}, \rho_{m+1}) = (\gamma \circ \mathcal{T}_m)(z'_m, \rho'_m)$. Since $\gamma \circ \mathcal{T}_m$ is an invertible transformation with unit Jacobian and inverse $(\gamma \circ \mathcal{T}_m)^{-1} = \mathcal{T}_m \circ \gamma$, we get that

$$T_m(z_{m+1}, \rho_{m+1}|z_m, \rho_m) = T'_m((\mathcal{T}_m \circ \gamma)(z_{m+1}, \rho_{m+1})|z_m, \rho_m) \tag{20}$$

$$= s(\mathcal{T}_m^\rho(z_{m+1}, -\rho_{m+1})|\rho_m)\, g_a(\mathcal{T}_m^z(z_{m+1}, -\rho_{m+1}) - z_m), \tag{21}$$

where $\mathcal{T}_m^\rho$ is the operator that applies $\mathcal{T}_m$ and returns the second component of the result (and similarly for $\mathcal{T}_m^z$).

Now, we compute $U_m(z_m, \rho_m|z_{m+1}, \rho_{m+1})$. We have that $(z'_m, \rho'_m) = (\mathcal{T}_m \circ \gamma)(z_{m+1}, \rho_{m+1})$. Thus,

$$U_m(z_m, \rho_m|z_{m+1}, \rho_{m+1}) = U_m(z_m, \rho_m|z'_m, \rho'_m) \tag{22}$$

$$= s_{\text{rev}}(\rho_m|\rho'_m)\, g_a(z_m - z'_m) \tag{23}$$

$$= s_{\text{rev}}(\rho_m|\mathcal{T}_m^\rho(z_{m+1}, -\rho_{m+1})\, g_a(z_m - \mathcal{T}_m^z(z_{m+1}, -\rho_{m+1})). \tag{24}$$

Taking the ratio $U_m(z_m, \rho_m|z_{m+1}, \rho_{m+1})/T_m(z_{m+1}, \rho_{m+1}|z_m, \rho_m)$ the factors involving the Gaussian pdf cancel (the density of a mean zero Gaussian is symmetric) and using that

$$s_{\text{rev}}(\rho_m|\rho'_m)S(\rho'_m) = s(\rho'_m|\rho_m)S(\rho_m) \longrightarrow \frac{s_{\text{rev}}(\rho_m|\rho'_m)}{s(\rho'_m|\rho_m)} = \frac{S(\rho_m)}{S(\rho'_m)} \tag{25}$$

yields get the desired result.