# OpenReview forum: "MCMC Variational Inference via Uncorrected Hamiltonian Annealing"
_NeurIPS.cc/2021/Conference — NeurIPS 2021 Poster_

### Official Review · Reviewer_5mVb · 2021-07-05

**Rating:** 8
**Confidence:** 4

**Summary:**

The authors introduce a variational inference method that leverages Annealed Importance Sampling (AIS) and Langevin/HMC MCMC. In particular HMC dynamics are used to target a sequence of intermediate bridging densities in order to generate approximate samples from the target posterior distribution. Importantly, since the method is uncorrected (i.e. does without a accept/reject step) the resulting lower bound is fully differentiable. This makes various hard-to-set parameters (e.g. temperatures for bridging densities) learnable. In experiments the authors demonstrate that their method, Uncorrected Hamiltonian Annealing (UHA), leads to tight lower bounds on the log evidence.

**Limitations And Societal Impact:**

The authors should more explicitly address some of the (potential) limitations of their work. For example: computational complexity, performance on multi-modal problems, non-applicability to discrete latent variables, etc

The authors do not address potential negative societal impact of their work.

**Main Review:**

This work presents an elegant approach for combining AIS and HMC within the confines of variational inference. While various authors have proposed methods that marry MCMC with variational inference over the years, some of the resulting constructions are somewhat clunky, requiring e.g. learning reverse models. In contrast the proposed method, UHA, is comparatively simple and does not require learning complex neural networks or the like. What is particularly attractive about this approach is that it effectively defines an implicit/semi-parametric variational distribution that makes extensive use of the target's gradient information. For this reason one would expect UHA to be an effective inference algorithm in a wide variety of scenarios without the need for model specific modifications (at least if the method is reasonably numerical stable). So while this work is in some sense incremental in that it combines existing algorithms from the literature, it does so in a principled way that cleverly addresses an important class of problems. The experiments are reasonably comprehensive and convincing.

Detailed questions and comments:

- The related work section is rather cursory. I think it would be very valuable if the authors extended the discussion there to give a more comprehensive comparison to existing methods that combine MCMC with variational inference. Even if there is not sufficient room in the main text, this discussion could be included in the supplemental materials. Apart from this shortcoming, however, the paper is generally clearly written.

- "In all cases, for UHA and Hamiltonian AIS we use under-damped Langevin dynamics, that is, we perform just one leapfrog step per transition and partially resample momentum." Can you comment on the choice of one leapfrog step? For a fixed amount of computation cost is the regime with more leapfrog steps and fewer bridging densities less effective? Or is this choice driven by numerical stabilities or some other concern?

- Can you comment on using different momentum distributions for each bridging density? It seems like this could be important. Is there a simple parametric scaling that suggests itself?

- Can you comment on numerical stability of the method? It seems you initialize with a vanilla mean field distribution obtained from ELBO maximization. Does everything break down without this initialization scheme, e.g. if you just initialize q(z) to be the unit normal distribution?

- How well can UHA deal with multi-modality? It would be informative to e.g. consider a 2-dimensional multi-modal example and depict the learned variational distribution for increasing values of K.

- "However, for smaller K, IW sometimes had better log-likelihoods than UHA (despite worse ELBOs)" Any hypothesis as to why this is?

- There is no discussion of runtime or computational complexity. It would be useful to get an idea of how expensive UHA becomes for large K.

- "For K ≥ 64 we tune the UHA’s parameters using K = 64 and extrapolate them as explained in Appendix D." Is this entirely because of computational cost or are there other factors in play here as well (e.g. stability)?

## After author response

I find that the authors have answered questions raised by the reviewers adequately and continue to believe that this paper should be accepted for publication. The elegance and (relative) simplicity of UHA make it a valuable contribution to the literature, and I imagine it is quite likely that follow-up work will exploit the ideas introduced here in other fruitful contexts.

**Time Spent Reviewing:**

1.25

---

> ### Author Response · Authors · 2021-08-09
> **Thank you for reviewing our work**
>
> Thank you for carefully reviewing our work and for the useful comments, suggestions and questions.
>
> 1. *The related work section is rather cursory. I think it would be very valuable if the authors extended the discussion[...]*
>
> We'd be happy to extend it.
>
> 2. *"In all cases, for UHA and Hamiltonian AIS [...] we perform one leapfrog step per transition and partially resample momentum." Can you comment on the choice of one leapfrog step? For a fixed amount of computation cost is the regime with more leapfrog steps and fewer bridging densities less effective? Or is this choice driven by numerical stabilities or some other concern?*
>
> We chose this for simplicity, as there are already many other parameters that are being trained/modified. Changing the number of leapfrog steps/bridging densities does not have a negative effect on UHA's performance. We tested this, and found that similar results were obtained with different choices that kept the number of likelihood evaluations constant (e.g. 64 bridging densities and 1 leapfrog step, 32 bridging densities and 2 leapfrog steps, 16 and 4). We'd be happy to include some additional results in the paper results showing this.
>
> 3. *Can you comment on using different momentum distributions for each bridging density? It seems like this could be important. Is there a simple parametric scaling that suggests itself?*
>
> While we did not explore this, we also think that it may yield further improvements. A simple option to try could be, for instance, setting the diagonal Gaussian's covariance as an affine function of $\beta$ (similar to what we tested for the bridging densities). This way the momentum distribution would gradually change as the bridging density moves away from the initial distribution and closer to the target. While there are more complex alternatives (e.g. use a neural network), we think this would be a good first step in exploring this.
>
> 4. *Can you comment on numerical stability of the method? It seems you initialize with a vanilla mean field distribution obtained from ELBO maximization. Does everything break down without this initialization scheme, e.g. if you just initialize q(z) to be the unit normal distribution?*
>
> The primary reason for using this initialization is to make our baseline method (HAIS) as strong as possible, since this is the only way we can tune its base distribution. Using a different initialization does not break any of the methods. Regarding stability, for the inference tasks we did not observe any issues. We only observe some stability issues for the VAE setting, which we addressed by constraining the leapfrog's step-size, see lines 282 and 305 (as noted in the original HVAE paper by Catherini et al., this may help prevent the leapfrog discretization from entering unstable regimes.)
>
> 5. *How well can UHA deal with multi-modality? It would be informative to e.g. consider a 2-dimensional multi-modal example and depict the learned variational distribution for increasing values of K.*
>
> We expect UHA to be neither better nor worse than Hamiltonian AIS for multi-modal distributions. We performed some additional simulations with a simple 2 dimensional bi-modal distribution to verify this empirically. As for Hamiltonian AIS, results depend on initialization. If the initial distribution is initialized to be concentrated around one of the modes, then the second mode will likely not be discovered (if no strong overlap). On the other hand, if the initial distribution is initialized covering both modes, then UHA recovers both modes correctly as $K$ increases. We can add these results to the paper.
>
> 6. *"However, for smaller K, IW sometimes had better log-likelihoods than UHA (despite worse ELBOs)" Any hypothesis as to why this is?*
>
> We have two hypothesis for this. First, generalization issues. For the architecture with two hidden layers we observed a larger gap between the ELBO for the training set and test set. Second, the ELBO in this work is actually an augmented ELBO; there may be more looseness in the augmented part with IW than there is with UHA. As mentioned above, these are hypothesis, and a definitive answer would require a more careful study.
>
> 7. *There is no discussion of runtime or computational complexity. It would be useful to get an idea of how expensive UHA becomes for large K.*
>
> All methods considered have $\mathcal{O}(K)$ cost (unless we extrapolate parameters for UHA, see next comment). In practice, for a fixed $K$, all HMC-based methods (UHA, HIS, HVAE) have similar time-cost, and tend to be slower than IW, with the difference depending on the model, $K$, implementation and computational platform. Just to give a rough idea, we ran optimization for 500 steps (no GPU) for $K = 8$ and $K = 32$ for the Brownian motion model and logistic regression with a1a. For Brownian motion and $K=8$ IW was around $\times 1.4$ times faster, and $\times 1.8$ for $K=32$. For logistic regression and $K=8$ IW was $\times 1.9$ times faster and $\times 3$ for $K=32$. (These numbers were obtained running things on a laptop without, and are expected to change using more computing power and/or a GPU.)
>
> 8. *"For $K \geq 64$ we tune the UHA’s parameters using K = 64 and extrapolate them" Is this entirely because of computational cost or are there other factors in play here as well (e.g. stability)?*
>
> This is due to the computational cost, which grows linearly with $K$. Tuning parameters for $K > 64$ directly does not have a negative effect on UHA's performance. However, we think that the extrapolation explained in the paper may be useful in practice if an extremely large $K$ is used. Something similar has been done quite regularly for IW in previous work (e.g. train using IW with $K=50$ and evaluate with a much larger $K$).
>
> 9. *The authors should more explicitly address some of the (potential) limitations of their work. For example: computational complexity, performance on multi-modal problems, non-applicability to discrete latent variables, etc.*
>
> We'd be happy to include a discussion regarding these.

---

### Official Review · Reviewer_BV2F · 2021-07-15

**Rating:** 8
**Confidence:** 2

**Summary:**

This paper studies the annealed important sampling (AIS) using the Hamiltonian dynamics kernels in each interpolant. To circumvent the problem that the Hamiltonian dynamics involves the non-differentiable operations, this paper proposes to discard the accept-reject operations in the kernel, and derives the resulting density ratios in a simple form related to the momentum variables. The resulting approach, named Uncorrected Hamiltonian Annealing (UHA), is a fully differentiable AIS method. This paper validates the effectiveness of UHA through extensive experiments and highlights the importance of differentiability.

**Limitations And Societal Impact:**

I do not see a big limitation of the approach.

**Main Review:**

## Novelty and Significance
Variational inference and MCMC are mainstream approaches for posterior inference, with both merits and shortcomings. Using the MCMC kernels in the variational posterior is an appealing direction to combine both merits together. And this paper makes an important step. By discarding the accept-reject step in the Hamiltonian kernel, UHA enables full differentiability. Compared to the standard AIS where the density ratios can be computed by the interpolated distributions, this paper presents an expression that can be simply obtained from the momentum variables in the Hamiltonian dynamics. Therefore, I think the proposed UHA is novel and  important in the field.

## The tightness of the bound
By using more samples, IWAE [1] provably tighten the lower bound compared to the standard ELBO, which enables better model learning. In contrast, the UHA introduces an augmented space for the inference. Is there any guarantees on the tightness of the bound ?

## Test-Time Encoder
Does the encoder at test-time also involve the UHA procedure ? If so, how large is the computational overhead compared to a standard NN encoder ?


## Clarity
This paper is well written. This paper clearly introduces the background knowledge and presents the improvement of the proposed approach. Detailed algorithm details are provided.

## Experiment
I do not have the expertise to fully evaluate the experiments. But I particularly enjoy seeing the improvements by turning more parameters in Figure 2 and Figure 3. I think they are strong evidences on the benefits of the proposed approach.

**Time Spent Reviewing:**

3

---

> ### Author Response · Authors · 2021-08-09
> **Thanks you for reviewing our work**
>
> Thank you for carefully reviewing our work and for the questions. We address them next.
>
> 1. *By using more samples IWAE provably tighten the bound [...] UHA introduces an augmented space for the inference. Is there any guarantees on the tightness of the bound?*
>
> This paper is premised on the idea that the bound will improve if AIS/UHA provides a better estimator than pure importance sampling, but there is no guarantee that this will be true for any particular problem. For context, our method reduces to traditional VI for $K=1$ because AIS/UHA reduces to importance sampling with a single sample for $K=1$ (for larger $K$ UHA also reduces to plain VI if we set $(\epsilon, \eta) = (0, 0)$).
>
> 2. *Does the encoder at test-time also involve the UHA procedure? If so, how large is the computational overhead compared to a standard NN encoder?*
>
> Depends on what is being measured at test-time. For VAEs, if we are estimating the marginal likelihood, then we run AIS with a very large number of bridging densities (see line 288 and table 2), which has exactly the same cost for all methods (IW, UHA, plain VAE training). In this case this does not involve the UHA procedure. The benefit of UHA is that during training, its tighter bound may lead to better models (decoders). If we are computing the ELBO, then we do run the UHA augmentation at test time (similar to IW, which computes the $K$-samples IW-bound). When computing ELBOs, for both IW and UHA the cost grows linearly with $K$.

---

### Official Review · Reviewer_uWL9 · 2021-07-15

**Rating:** 8
**Confidence:** 4

**Summary:**

The paper propose to use uncorrected HMC kernels (i.e. HMC kernels without accept-reject steps) in AIS to make the estimate of log normalisation constant differentiable w.r.t. to various parameters (e.g. step-size for numerical integration).
The author(s) name(s) the method UHA.
The promise is that one can then using reparameterisation gradient to tune these parameters.
Through experiments on inference tasks,
the benefit of tuning parameters in UHA is demonstrated against other methods that use HMC in AIS.
For experiments on VAE training,
UHA with tuned parameters has shown better test ELBO than IW.

**Limitations And Societal Impact:**

Limitations are discussed in section 6.
There is no societal impact discussed but I don't think it's needed for this technical paper.

**Main Review:**

Originality:
UHA is a novel method derived by a neat modification to the common way of using HMC in AIS.
It solves an important limitation of previous methods: they are sensitive to hyper-parameters but there is no easy way to tune them.
Differences against previous methods are made clear through the motivation as well as the dedicated related work section.

Quality:
The method is technically sound and the benefit of being able to tuning all parameters is clearly demonstrated by experiments in section 5.1 and 5.2.

Clarity:
The paper is very well-written and easy to follow.
Technical details are also enough for reproducing the results.

Significance:
The fact that the proposed method leads to a differentiable estimate is a solid contribution to the community.
One of the biggest challenges to use HMC in AIS is about hyper-parameter tuning and UHA and I can see how UHA would popularise the use of AIS as now the hyper-parameter tuning is made easy.

Misc:
- An interesting baseline that is not included is tuning parameters using the REINFORCE estimator (either in UHA or HAIS), although I could expect it would be worse than the reparameterisation estimator.
- The grid search for HAIS is insufficient. One interesting thing to check from a more fine-grained search of HAIS is that we could see how much we lose by getting rid of the accept-reject step. Another interesting thing to check is how different the optimal parameters are for HAIS and UHA.
- For the last sentence from L291, how do we know the true $\log Z$ to compute the variational gap?

**Time Spent Reviewing:**

1.5

---

> ### Author Response · Authors · 2021-08-09
> **Thank you for reviewing our work**
>
> Thank you for carefully reviewing our work and for the useful comments suggestions. We address each question next.
>
> 1. *A baseline that is not included is tuning parameters using the REINFORCE estimator (either in UHA or HAIS), although I could expect it would be worse than the reparameterization estimator.*
>
> We believe this would be most sensible for Hamiltonian AIS. While we don't expect this baseline to work well, we could include it in some experiments if it may improve the paper.
>
> 2. *The grid search for HAIS is insufficient. One interesting thing to check [...] how much we lose by getting rid of the accept-reject step. Another interesting thing [...] how different the optimal parameters are for HAIS and UHA.*
>
> We did some simulations (not present in the paper) comparing UHA tuning $(\epsilon, \eta)$ against HAIS tuning the same two parameters by grid search using a very fine grid ($2500$ different pairs $(\epsilon, \eta)$). In short, while using a finer grid may lead to slightly better performances for HAIS, there is no qualitative change. In any case, we could add these results in the paper if they are considered informative.
>
> 3. *For the last sentence from L291, how do we know the true log Z to compute the variational gap?*
>
> We estimate it using AIS with HMC and a very large number of bridging densities (see lines 288 and caption for table 2).

---

> > ### Comment · Reviewer_uWL9 · 2021-08-16
> > **Thanks for the author reponse**
> >
> > Thanks for the reponses and I will keep my score here.
> >
> > > We did some simulations (not present in the paper) ...
> >
> > It would be nice to put that fine-grained simulation in the appendix.

---

### Official Review · Reviewer_x5iz · 2021-07-16

**Rating:** 7
**Confidence:** 3

**Summary:**

The paper proposes UHA, an original and clever modification of Hamiltonian AIS. The idea is to change the AIS ratio such that it becomes tunable, but can still be computed in closed form (ratios of momentum distributions). This is achieved by dropping the accept-reject step in the HMC update such that a single UHA step only involves (re)sampling momenta and leapfrog integration. This modification allows the authors to tune parameters such as the proposal distribution, the step size and damping coefficients (used in momentum resampling). Many interesting ideas for future extensions are proposed. Experiments on various models demonstrate that UHA obtains a tighter ELBO than other combinations of VI and HMC.


**Limitations And Societal Impact:**

-

**Main Review:**

# Contributions

* Variational inference with Uncorrected Hamiltonian Annealing (UHA)

* Achieves tighter ELBO and allows optimization of parameters

# Originality

The paper proposes UHA, an original and clever modification of Hamiltonian AIS. The idea is to change the AIS ratio such that it becomes tunable, but can still be computed in closed form (ratios of momentum distributions). This is achieved by dropping the accept-reject step in the HMC update such that a single UHA step only involves (re)sampling momenta and leapfrog integration. This modification allows the authors to tune parameters such as the proposal distribution, the step size and damping coefficients (used in momentum resampling). Many interesting ideas for future extensions are proposed.

# Quality

The paper is technically sound and the experiments are instructive.

# Clarity

In general, the paper is well written.

Although the notation is generally consistent, there are a few ambiguities:

- Page 2, Lines 47/48: Here you denote the number of bridging distributions by $K$ although in the remainder $M$ is the number of bridging distributions (and $K$ the number of likelihood evaluations)

- Page 5, Line 172: How are the $\beta_m$ chosen? When tuning also the $\beta_m$ (Figure 2), it would be interesting to see the tuned annealing schedule.

- Page 6, Lines 210-213: What is $d$? I guess the dimension of $z$ space?

# Significance

The experiments show that UHA outperforms other variants of Hamiltonian sampling and variational inference.

However, to better judge the improvements achieved with UHA it would be useful to also show the correct log evidence (or rather an accurate estimate obtained, for example, with standard AIS using many bridging distributions and multiple parallel trajectories). It would be helpful to show these estimates in Figure 1-3.

# Comments

* Line 204: You perform only a single leapfrog step. Have you run tested with more leapfrog steps?

* In standard AIS, multiple annealing sequences ("trajectories") are simulated independently. Have you considered doing something similar with UHA (resulting in yet another augmentation of the sample space)?


**Time Spent Reviewing:**

4

---

> ### Author Response · Authors · 2021-08-09
> **Thank you for reviewing our work**
>
> Thank you for carefully reviewing our work and for the useful comments suggestions. We address each question next.
>
> 1. *How are the $\beta_m$ chosen? When tuning $\beta_m$ (Figure 2), it would be interesting to see the tuned annealing schedule.*
>
> By default, the values for the $\beta$ parameters are chosen to be linearly spaced between zero and one. When these parameters are tuned, they are initialized that way. We'd be happy to add a plot for a few of the models from Section 5.1.1 showing how the tuned parameters $\beta_m$ look like.
>
> 2. *What is $d$? I guess the dimension of $z$ space?*
>
> Yes, we'll clarify this in the paper.
>
> 3. *To better judge the improvements achieved with UHA it would be useful to show the correct log evidence (or accurate estimate obtained with AIS with many bridging densities and parallel trajectories). It would be helpful to show these estimates in Figure 1-3.*
>
> We agree, this would be useful when analyzing results. Following this comment, we estimated the log evidence for each of the four models using Hamiltonian AIS with 1500 bridging densities, 16 leapfrog steps per bridging density, $\eta = 0.8$, and a step-size $\epsilon$ tuned to achieve a rejection rate of $0.05$. The results are: $-677.3$ for logistic regression with the a1a dataset, $-2398.6$ for the madelon dataset, $1.1$ for the Brownian motion model, and $-1151$ for the Lorenz convection model. We'll add these results to the paper. As it can be observed in Figure 2 in the paper, UHA tuning all parameters gets quite close to these values.
>
> 4. *You perform only a single leapfrog step. Have you run tested with more leapfrog steps?*
>
> We did preliminary tests using a different number of leapfrog steps per bridging density (not present in the paper). Overall, we found that different choices that keep the number of likelihood evaluations constant (e.g. 64 bridging densities with 1 leapfrog step, 32 bridging densities with 2 leapfrog steps, or 16 bridging densities with 4 leapfrog steps) yield similar results. We decided to show results only involving one leapfrog step for simplicity and clarity, since many other parameters are already being modified. We could add some results using different number of leapfrog steps in the final version of the paper.
>
> 5. *In standard AIS, multiple annealing sequences ("trajectories") are simulated independently. Have you considered doing something similar with UHA (resulting in yet another augmentation of the sample space)?*
>
> Is the question about combining UHA with IW? This is possible. We considered it, but decided not to pursue it for simplicity and clarity.
>
> On the other hand, the UHA bound shown in all plots is estimated using several samples (same for all other methods, also typically done for AIS as mentioned in the review).

---

### Official Review · Reviewer_kH8D · 2021-08-16

**Rating:** 7
**Confidence:** 4

**Summary:**

The paper proposed to use the uncorrected HMC kernel in an AIS-type variational scheme. This enables optimizing the ELBO via reparameterization gradients for many algorithm tuning parameters, including step size, momentum covariance, and annealing schedule. The method is shown to outperform competitors in most cases.

**Limitations And Societal Impact:**

There was some discussion of limitations

**Main Review:**

The proposed method appears to be sound and potentially quite useful. A few suggestions:

1. The proof of Thm 2 lacks some rigor – it would be better to argue measure theoretically.

2. Many of the papers cited as being on arXiv are published either in workshops or conferences. The author(s) should check all arXiv citations carefully. One tricky case is [17], where there's a related published paper https://papers.nips.cc/paper/2016/file/0e9fa1f3e9e66792401a6972d477dcc3-Paper.pdf

3. There are a few missing pieces and areas for improvement in the experiments:

- Please include training runtimes for the different methods
- I would have liked to see a "toy" example with known log Z to demonstrate the tightness of the lower bounds produced by the proposed method
- Given that the proposed method produces a posterior approximation, some comparison of the approximation accuracy compared to, say, Stan's dynamic HMC implementation would have been very interesting. I realize posterior approximation per se is not the focus of the paper, but I wonder how competitive the proposed method would be with dynamic HMC when measured in terms of number of gradient evaluations. Even if it were not quite as good, the fact that is also produces an estimate of the marginal likelihood could make the method a valuable alternative when the user wants to do model selection

**Time Spent Reviewing:**

1.5

---

> ### Author Response · Authors · 2021-08-21
> **Thank you for reviewing our work**
>
> Thank you for carefully reviewing our work and for the great suggestions.
>
> 1. *Citations to arxiv version instead of published papers.*
>
> We apologize for this. We'll fix all these citation issues. For the tricky cases, as [17], we will include citations to both versions.
>
> 2. *Toy example and running times.*
>
> We think this is a good suggestion. Following this, we propose to use as toy model a Student-t distribution with dimension 500, location 0, scale 1, and 3 degrees of freedom. Since the distribution is normalized we know that $\log Z = 0$. We take the base distribution to be Gaussian. We tested plain VI, IW for $K \in$\{$128, 1024$\} and UHA for $K \in$\{$4, 16, 64, 128$\}. The lower bounds achieved in each case are:
>
> - Plain VI: $-20.4$.
> - UHA ($K=4$): $-13.8$.
> - UHA ($K=16$): $-8.9$.
> - UHA ($K=64$): $-4.9$.
> - UHA ($K=128$): $-3.6$.
> - IW ($K=128$): $-11.9$.
> - IW ($K=1024$): $-10.4$.
>
> In this problem, UHA with $K=16$ performs better than IW with $K=1024$. We can include these results in the paper. (And also for other dimensionalities.) We also include the running times for 500 optimization steps for this experiment:
>
> - Plain VI: $0.27$ secs.
> - UHA ($K=4$): $0.32$ secs.
> - UHA ($K=16$): $0.42$ secs.
> - UHA ($K=64$): $0.96$ secs.
> - UHA ($K=128$): $1.4$ secs.
> - IW ($K=128$): $1.3$ secs.
> - IW ($K=1024$): $4.5$ secs.
>
> In this case UHA and IW have very similar running times for the same $K$. In other cases UHA tends to be slower (see answer to reviewer 5mVb - item 7 in our response). We'll add running times for some of the experiments in the paper.
>
> 3. *Comparison against dynamic HMC (e.g. Stan)*
>
> We think this is a good suggestion too, and we are planning to include some results along this line in the paper. For simplicity, we could do this for a few of the models we already have. We could give a budget of $B$ gradient evaluations to both methods, and use part of them for training (burn-in in the case of Stan) and the remainder for evaluation. As for measuring approximation accuracy, we are planning on measuring moment errors. That is, get the true 1st and 2nd order moment of the target (if known, otherwise estimate using AIS with a large number of bridging densities or HMC ran for a very long time), get the moments obtained by each of the methods using the evaluation budget, and measure the difference.

---

> > ### Comment · Reviewer_kH8D · 2021-08-31
> > **thanks**
> >
> > Thank you for addressing my concerns. I think these additional experiments will really strengthen the paper.

---

### Decision · Program_Chairs · 2021-09-27

**Decision:**

Accept (Poster)

**Comment:**

The reviewers all came to a clear consensus that this paper should be accepted. The authors addressed all points raised by reviewers thoroughly. In the camera ready, please make sure to follow through on the edits discussed during the review process. In particular, multiple reviewers pointed out that the experiments should include runtimes, and that there should also be an experiment that provides a comparison to the true log marginal -- please ensure these are included in the camera-ready. Please also improve the rigour of the proof of Theorem 2 using measure-theoretic arguments.